# Soft-to-Hard Vector Quantization for End-to-End Learning Compressible Representations

**Eirikur Agustsson**
ETH Zurich
aeirikur@vision.ee.ethz.ch

**Fabian Mentzer**
ETH Zurich
mentzerf@vision.ee.ethz.ch

**Michael Tschannen**
ETH Zurich
michaelt@nari.ee.ethz.ch

**Lukas Cavigelli**
ETH Zurich
cavigelli@iis.ee.ethz.ch

**Radu Timofte**
ETH Zurich & Merantix
timofter@vision.ee.ethz.ch

**Luca Benini**
ETH Zurich
benini@iis.ee.ethz.ch

**Luc Van Gool**
KU Leuven & ETH Zurich
vangool@vision.ee.ethz.ch

## Abstract

We present a new approach to learn compressible representations in deep architectures with an end-to-end training strategy. Our method is based on a soft (continuous) relaxation of quantization and entropy, which we anneal to their discrete counterparts throughout training. We showcase this method for two challenging applications: Image compression and neural network compression. While these tasks have typically been approached with different methods, our soft-to-hard quantization approach gives results competitive with the state-of-the-art for both.

## 1 Introduction

In recent years, deep neural networks (DNNs) have led to many breakthrough results in machine learning and computer vision [20, 28, 10], and are now widely deployed in industry. Modern DNN models often have millions or tens of millions of parameters, leading to highly redundant structures, both in the intermediate feature representations they generate and in the model itself. Although overparametrization of DNN models can have a favorable effect on training, in practice it is often desirable to compress DNN models for inference, *e.g.*, when deploying them on mobile or embedded devices with limited memory. The ability to learn compressible feature representations, on the other hand, has a large potential for the development of (data-adaptive) compression algorithms for various data types such as images, audio, video, and text, for all of which various DNN architectures are now available.

DNN model compression and lossy image compression using DNNs have both independently attracted a lot of attention lately. In order to compress a set of continuous model parameters or features, we need to approximate each parameter or feature by one representative from a set of quantization levels (or vectors, in the multi-dimensional case), each associated with a symbol, and then store the assignments (symbols) of the parameters or features, as well as the quantization levels. Representing each parameter of a DNN model or each feature in a feature representation by the corresponding quantization level will come at the cost of a distortion $D$, *i.e.*, a loss in performance (*e.g.*, in classification accuracy for a classification DNN with quantized model parameters, or in reconstruction error in the context of autoencoders with quantized intermediate feature representations). The rate $R$, *i.e.*, the entropy of the symbol stream, determines the cost of encoding the model or features in a bitstream.

To learn a compressible DNN model or feature representation we need to minimize $D + \beta R$, where $\beta > 0$ controls the rate-distortion trade-off. Including the entropy into the learning cost function can be seen as adding a regularizer that promotes a compressible representation of the network or feature representation. However, two major challenges arise when minimizing $D + \beta R$ for DNNs: i) coping with the non-differentiability (due to quantization operations) of the cost function $D + \beta R$, and ii) obtaining an accurate and differentiable estimate of the entropy (*i.e.*, $R$). To tackle i), various methods have been proposed. Among the most popular ones are stochastic approximations [39, 19, 7, 32, 5] and rounding with a smooth derivative approximation [15, 30]. To address ii) a common approach is to assume the symbol stream to be i.i.d. and to model the marginal symbol distribution with a parametric model, such as a Gaussian mixture model [30, 34], a piecewise linear model [5], or a Bernoulli distribution [33] (in the case of binary symbols).

In this paper, we propose a unified end-to-end learning framework for learning compressible representations, jointly optimizing the model parameters, the quantization levels, and the entropy of the resulting symbol stream to compress either a subset of feature representations in the network or the model itself (see inset figure). We address both challenges i) and ii) above with methods that are novel in the context DNN model and feature compression. Our main contributions are:

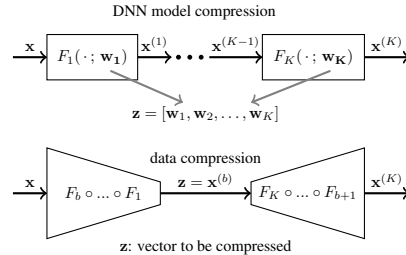

- We provide the first unified view on end-to-end learned compression of feature representations and DNN models. These two problems have been studied largely independently in the literature so far.

- Our method is simple and intuitively appealing, relying on soft assignments of a given scalar or vector to be quantized to quantization levels. A parameter controls the "hardness" of the assignments and allows to gradually transition from soft to hard assignments during training. In contrast to rounding-based or stochastic quantization schemes, our coding scheme is directly differentiable, thus trainable end-to-end.

- Our method does not force the network to adapt to specific (given) quantization outputs (*e.g.*, integers) but learns the quantization levels jointly with the weights, enabling application to a wider set of problems. In particular, we explore vector quantization for the first time in the context of learned compression and demonstrate its benefits over scalar quantization.

- Unlike essentially all previous works, we make no assumption on the marginal distribution of the features or model parameters to be quantized by relying on a histogram of the assignment probabilities rather than the parametric models commonly used in the literature.

- We apply our method to DNN model compression for a 32-layer ResNet model [13] and full-resolution image compression using a variant of the compressive autoencoder proposed recently in [30]. In both cases, we obtain performance competitive with the state-of-the-art, while making fewer model assumptions and significantly simplifying the training procedure compared to the original works [30, 6].

The remainder of the paper is organized as follows. Section 2 reviews related work, before our soft-to-hard vector quantization method is introduced in Section 3. Then we apply it to a compressive autoencoder for image compression and to ResNet for DNN compression in Section 4 and 5, respectively. Section 6 concludes the paper.

## 2   Related Work

There has been a surge of interest in DNN models for full-resolution image compression, most notably [32, 33, 4, 5, 30], all of which outperform JPEG [35] and some even JPEG 2000 [29] The pioneering work [32, 33] showed that progressive image compression can be learned with convolutional recurrent neural networks (RNNs), employing a stochastic quantization method during training. [4, 30] both rely on convolutional autoencoder architectures. These works are discussed in more detail in Section 4.

In the context of DNN model compression, the line of works [12, 11, 6] adopts a multi-step procedure in which the weights of a pretrained DNN are first pruned and the remaining parameters are quantized using a $k$-means like algorithm, the DNN is then retrained, and finally the quantized DNN model is encoded using entropy coding. A notable different approach is taken by [34], where the DNN

compression task is tackled using the minimum description length principle, which has a solid information-theoretic foundation.

It is worth noting that many recent works target quantization of the DNN model parameters and possibly the feature representation to speed up DNN evaluation on hardware with low-precision arithmetic, see, *e.g.*, [15, 23, 38, 43]. However, most of these works do not specifically train the DNN such that the quantized parameters are compressible in an information-theoretic sense.

Gradually moving from an easy (convex or differentiable) problem to the actual harder problem during optimization, as done in our soft-to-hard quantization framework, has been studied in various contexts and falls under the umbrella of continuation methods (see [3] for an overview). Formally related but motivated from a probabilistic perspective are deterministic annealing methods for maximum entropy clustering/vector quantization, see, *e.g.*, [24, 42]. Arguably most related to our approach is [41], which also employs continuation for nearest neighbor assignments, but in the context of learning a supervised prototype classifier. To the best of our knowledge, continuation methods have not been employed before in an end-to-end learning framework for neural network-based image compression or DNN compression.

# 3 Proposed Soft-to-Hard Vector Quantization

## 3.1 Problem Formulation

**Preliminaries and Notations.** We consider the standard model for DNNs, where we have an architecture $F : \mathbb{R}^{d_1} \mapsto \mathbb{R}^{d_{K+1}}$ composed of $K$ layers $F = F_K \circ \cdots \circ F_1$, where layer $F_i$ maps $\mathbb{R}^{d_i} \to \mathbb{R}^{d_{i+1}}$, and has parameters $\mathbf{w}_i \in \mathbb{R}^{m_i}$. We refer to $\mathbf{W} = [\mathbf{w}_1, \cdots, \mathbf{w}_K]$ as the parameters of the network and we denote the intermediate layer outputs of the network as $\mathbf{x}^{(0)} := \mathbf{x}$ and $\mathbf{x}^{(i)} := F_i(\mathbf{x}^{(i-1)})$, such that $F(\mathbf{x}) = \mathbf{x}^{(K)}$ and $\mathbf{x}^{(i)}$ is the feature vector produced by layer $F_i$.

The parameters of the network are learned w.r.t. training data $\mathcal{X} = \{\mathbf{x}_1, \cdots, \mathbf{x}_N\} \subset \mathbb{R}^{d_1}$ and labels $\mathcal{Y} = \{\mathbf{y}_1, \cdots, \mathbf{y}_N\} \subset \mathbb{R}^{d_{K+1}}$, by minimizing a real-valued loss $\mathcal{L}(\mathcal{X}, \mathcal{Y}; F)$. Typically, the loss can be decomposed as a sum over the training data plus a regularization term,

$$\mathcal{L}(\mathcal{X}, \mathcal{Y}; F) = \frac{1}{N} \sum_{i=1}^{N} \ell(F(\mathbf{x}_i), \mathbf{y}_i) + \lambda R(\mathbf{W}), \tag{1}$$

where $\ell(F(\mathbf{x}), \mathbf{y})$ is the sample loss, $\lambda > 0$ sets the regularization strength, and $R(\mathbf{W})$ is a regularizer (*e.g.*, $R(\mathbf{W}) = \sum_i \|\mathbf{w}_i\|^2$ for $l_2$ regularization). In this case, the parameters of the network can be learned using stochastic gradient descent over mini-batches. Assuming that the data $\mathcal{X}, \mathcal{Y}$ on which the network is trained is drawn from some distribution $P_{\mathsf{X},\mathsf{Y}}$, the loss (1) can be thought of as an estimator of the expected loss $\mathbb{E}[\ell(F(\mathsf{X}), \mathsf{Y}) + \lambda R(\mathbf{W})]$. In the context of image classification, $\mathbb{R}^{d_1}$ would correspond to the input image space and $\mathbb{R}^{d_{K+1}}$ to the classification probabilities, and $\ell$ would be the categorical cross entropy.

We say that the deep architecture is an autoencoder when the network maps back into the input space, with the goal of reproducing the input. In this case, $d_1 = d_{K+1}$ and $F(\mathbf{x})$ is trained to approximate $\mathbf{x}$, *e.g.*, with a mean squared error loss $\ell(F(\mathbf{x}), \mathbf{y}) = \|F(\mathbf{x}) - \mathbf{y}\|^2$. Autoencoders typically condense the dimensionality of the input into some smaller dimensionality inside the network, *i.e.*, the layer with the smallest output dimension, $\mathbf{x}^{(b)} \in \mathbb{R}^{d_b}$, has $d_b \ll d_1$, which we refer to as the "bottleneck".

**Compressible representations.** We say that a weight parameter $\mathbf{w}_i$ or a feature $\mathbf{x}^{(i)}$ has a compressible representation if it can be serialized to a binary stream using few bits. For DNN compression, we want the entire network parameters $\mathbf{W}$ to be compressible. For image compression via an autoencoder, we just need the features in the bottleneck, $\mathbf{x}^{(b)}$, to be compressible.

Suppose we want to compress a feature representation $\mathbf{z} \in \mathbb{R}^d$ in our network (*e.g.*, $\mathbf{x}^{(b)}$ of an autoencoder) given an input $\mathbf{x}$. Assuming that the data $\mathcal{X}, \mathcal{Y}$ is drawn from some distribution $P_{\mathsf{X},\mathsf{Y}}$, $\mathbf{z}$ will be a sample from a continuous random variable $\mathsf{Z}$.

To store $\mathbf{z}$ with a finite number of bits, we need to map it to a discrete space. Specifically, we map $\mathbf{z}$ to a sequence of $m$ symbols using a (symbol) encoder $E : \mathbb{R}^d \mapsto [L]^m$, where each symbol is an index ranging from 1 to $L$, *i.e.*, $[L] := \{1, \ldots, L\}$. The reconstruction of $\mathbf{z}$ is then produced by a (symbol) decoder $D : [L]^m \mapsto \mathbb{R}^d$, which maps the symbols back to $\hat{\mathbf{z}} = D(E(\mathbf{z})) \in \mathbb{R}^d$. Since $\mathbf{z}$ is

a sample from $\mathsf{Z}$, the symbol stream $E(\mathbf{z})$ is drawn from the discrete probability distribution $P_{E(\mathsf{Z})}$. Thus, given the encoder $E$, according to Shannon's source coding theorem [8], the correct metric for compressibility is the entropy of $E(\mathsf{Z})$:

$$H(E(\mathsf{Z})) = - \sum_{\mathbf{e} \in [L]^m} P(E(\mathsf{Z}) = \mathbf{e}) \log(P(E(\mathsf{Z}) = \mathbf{e})). \tag{2}$$

Our generic goal is hence to optimize the rate distortion trade-off between the expected loss and the entropy of $E(\mathsf{Z})$:

$$\min_{E,D,\mathbf{W}} \mathbb{E}_{\mathsf{X},\mathsf{Y}}[\ell(\hat{F}(\mathsf{X}),\mathsf{Y}) + \lambda R(\mathbf{W})] + \beta H(E(\mathsf{Z})), \tag{3}$$

where $\hat{F}$ is the architecture where $\mathbf{z}$ has been replaced with $\hat{\mathbf{z}}$, and $\beta > 0$ controls the trade-off between compressibility of $\mathbf{z}$ and the distortion it imposes on $\hat{F}$.

However, we cannot optimize (3) directly. First, we do not know the distribution of $\mathsf{X}$ and $\mathsf{Y}$. Second, the distribution of $\mathsf{Z}$ depends in a complex manner on the network parameters $\mathbf{W}$ and the distribution of $\mathsf{X}$. Third, the encoder $E$ is a discrete mapping and thus not differentiable. For our first approximation we consider the sample entropy instead of $H(E(\mathsf{Z}))$. That is, given the data $\mathcal{X}$ and some fixed network parameters $\mathbf{W}$, we can estimate the probabilities $P(E(\mathsf{Z}) = \mathbf{e})$ for $\mathbf{e} \in [L]^m$ via a histogram. For this estimate to be accurate, we however would need $|\mathcal{X}| \gg L^m$. If $\mathbf{z}$ is the bottleneck of an autoencoder, this would correspond to trying to learn a single histogram for the entire discretized data space. We relax this by assuming the entries of $E(\mathsf{Z})$ are i.i.d. such that we can instead compute the histogram over the $L$ distinct values. More precisely, we assume that for $\mathbf{e} = (e_1, \cdots, e_m) \in [L]^m$ we can approximate $P(E(\mathsf{Z}) = \mathbf{e}) \approx \prod_{l=1}^{m} p_{e_l}$, where $p_j$ is the histogram estimate

$$p_j := \frac{|\{e_l(\mathbf{z}_i) | l \in [m], i \in [N], e_l(\mathbf{z}_i) = j\}|}{mN}, \tag{4}$$

where we denote the entries of $E(\mathbf{z}) = (e_1(\mathbf{z}), \cdots, e_m(\mathbf{z}))$ and $\mathbf{z}_i$ is the output feature $\mathbf{z}$ for training data point $\mathbf{x}_i \in \mathcal{X}$. We then obtain an estimate of the entropy of $\mathsf{Z}$ by substituting the approximation (3.1) into (2),

$$H(E(\mathsf{Z})) \approx - \sum_{\mathbf{e} \in [L]^m} \left( \prod_{l=1}^{m} p_{e_l} \right) \log \left( \prod_{l=1}^{m} p_{e_l} \right) = -m \sum_{j=1}^{L} p_j \log p_j = mH(p), \tag{5}$$

where the first (exact) equality is due to [8], Thm. 2.6.6, and $H(p) := - \sum_{j=1}^{L} p_j \log p_j$ is the sample entropy for the (i.i.d., by assumption) components of $E(\mathsf{Z})$ [1].

We now can simplify the ideal objective of (3), by replacing the expected loss with the sample mean over $\ell$ and the entropy using the sample entropy $H(p)$, obtaining

$$\frac{1}{N} \sum_{i=1}^{N} \ell(F(\mathbf{x}_i), \mathbf{y}_i) + \lambda R(\mathbf{W}) + \beta m H(p). \tag{6}$$

We note that so far we have assumed that $\mathbf{z}$ is a feature output in $F$, i.e., $\mathbf{z} = \mathbf{x}^{(k)}$ for some $k \in [K]$. However, the above treatment would stay the same if $\mathbf{z}$ is the concatenation of multiple feature outputs. One can also obtain a separate sample entropy term for separate feature outputs and add them to the objective in (6).

In case $\mathbf{z}$ is composed of one or more parameter vectors, such as in DNN compression where $\mathbf{z} = \mathbf{W}$, $\mathbf{z}$ and $\hat{\mathbf{z}}$ cease to be random variables, since $\mathbf{W}$ is a parameter of the model. That is, opposed to the case where we have a source $\mathcal{X}$ that produces another source $\hat{\mathsf{Z}}$ which we want to be compressible, we want the discretization of a single parameter vector $\mathbf{W}$ to be compressible. This is analogous to compressing a single document, instead of learning a model that can compress a stream of documents. In this case, (3) is not the appropriate objective, but our simplified objective in (6) remains appropriate. This is because a standard technique in compression is to build a statistical model of the (finite) data, which has a small sample entropy. The only difference is that now the histogram probabilities in (4) are taken over $\mathbf{W}$ instead of the dataset $\mathcal{X}$, i.e., $N = 1$ and $\mathbf{z}_i = \mathbf{W}$ in (4), and they count towards storage as well as the encoder $E$ and decoder $D$.

**Challenges.** Eq. (6) gives us a unified objective that can well describe the trade-off between compressible representations in a deep architecture and the original training objective of the architecture.

However, the problem of finding a good encoder $E$, a corresponding decoder $D$, and parameters $\mathbf{W}$ that minimize the objective remains. First, we need to impose a form for the encoder and decoder, and second we need an approach that can optimize (6) w.r.t. the parameters $\mathbf{W}$. Independently of the choice of $E$, (6) is challenging since $E$ is a mapping to a finite set and, therefore, not differentiable. This implies that neither $H(p)$ is differentiable nor $\hat{F}$ is differentiable w.r.t. the parameters of $\mathbf{z}$ and layers that feed into $\mathbf{z}$. For example, if $\hat{F}$ is an autoencoder and $\mathbf{z} = \mathbf{x}^{(b)}$, the output of the network will not be differentiable w.r.t. $\mathbf{w}_1, \cdots, \mathbf{w}_b$ and $\mathbf{x}^{(0)}, \cdots, \mathbf{x}^{(b-1)}$.

These challenges motivate the design decisions of our soft-to-hard annealing approach, described in the next section.

## 3.2 Our Method

**Encoder and decoder form.** For the encoder $E : \mathbb{R}^d \mapsto [L]^m$ we assume that we have $L$ centers vectors $\mathcal{C} = \{\mathbf{c}_1, \cdots, \mathbf{c}_L\} \subset \mathbb{R}^{d/m}$. The encoding of $\mathbf{z} \in R^d$ is then performed by reshaping it into a matrix $\mathbf{Z} = [\bar{\mathbf{z}}^{(1)}, \cdots, \bar{\mathbf{z}}^{(m)}] \in \mathbb{R}^{(d/m) \times m}$, and assigning each column $\bar{\mathbf{z}}^{(l)}$ to the index of its nearest neighbor in $\mathcal{C}$. That is, we assume the feature $\mathbf{z} \in \mathbb{R}^d$ can be modeled as a sequence of $m$ points in $\mathbb{R}^{d/m}$, which we partition into the Voronoi tessellation over the centers $\mathcal{C}$. The decoder $D : [L]^m \mapsto \mathbb{R}^d$ then simply constructs $\hat{\mathbf{Z}} \in \mathbb{R}^{(d/m) \times m}$ from a symbol sequence $(e_1, \cdots, e_m)$ by picking the corresponding centers $\hat{\mathbf{Z}} = [\mathbf{c}_{e_1}, \cdots, \mathbf{c}_{e_m}]$, from which $\hat{z}$ is formed by reshaping $\hat{\mathbf{Z}}$ back into $\mathbb{R}^d$. We will interchangeably write $\hat{\mathbf{z}} = D(E(\mathbf{z}))$ and $\hat{\mathbf{Z}} = D(E(\mathbf{Z}))$.

The idea is then to relax $E$ and $D$ into continuous mappings via soft assignments instead of the hard nearest neighbor assignment of $E$.

**Soft assignments.** We define the soft assignment of $\bar{\mathbf{z}} \in \mathbb{R}^{d/m}$ to $\mathcal{C}$ as

$$\phi(\bar{\mathbf{z}}) := \text{softmax}(-\sigma[\|\bar{\mathbf{z}} - \mathbf{c}_1\|^2, \ldots, \|\bar{\mathbf{z}} - \mathbf{c}_L\|^2]) \in \mathbb{R}^L, \tag{7}$$

where $\text{softmax}(y_1, \cdots, y_L)_j := \frac{e^{y_j}}{e^{y_1} + \cdots + e^{y_L}}$ is the standard softmax operator, such that $\phi(\bar{\mathbf{z}})$ has positive entries and $\|\phi(\bar{\mathbf{z}})\|_1 = 1$. We denote the $j$-th entry of $\phi(\bar{\mathbf{z}})$ with $\phi_j(\bar{\mathbf{z}})$ and note that

$$\lim_{\sigma \to \infty} \phi_j(\bar{\mathbf{z}}) = \begin{cases} 1 & \text{if } j = \arg\min_{j' \in [L]} \|\bar{\mathbf{z}} - \mathbf{c}_{j'}\| \\ 0 & \text{otherwise} \end{cases}$$

such that $\hat{\phi}(\bar{\mathbf{z}}) := \lim_{\sigma \to \infty} \phi(\bar{\mathbf{z}})$ converges to a one-hot encoding of the nearest center to $\bar{\mathbf{z}}$ in $\mathcal{C}$. We therefore refer to $\hat{\phi}(\bar{\mathbf{z}})$ as the hard assignment of $\bar{\mathbf{z}}$ to $\mathcal{C}$ and the parameter $\sigma > 0$ as the hardness of the soft assignment $\phi(\bar{\mathbf{z}})$.

Using soft assignment, we define the soft quantization of $\bar{\mathbf{z}}$ as

$$\tilde{Q}(\bar{\mathbf{z}}) := \sum_{j=1}^{L} \mathbf{c}_j \phi_i(\bar{\mathbf{z}}) = \mathbf{C}\phi(\bar{\mathbf{z}}),$$

where we write the centers as a matrix $\mathbf{C} = [\mathbf{c}_1, \cdots, \mathbf{c}_L] \in \mathbb{R}^{d/m \times L}$. The corresponding hard assignment is taken with $\hat{Q}(\bar{\mathbf{z}}) := \lim_{\sigma \to \infty} \tilde{Q}(\bar{\mathbf{z}}) = \mathbf{c}_{e(\bar{\mathbf{z}})}$, where $e(\bar{\mathbf{z}})$ is the center in $\mathcal{C}$ nearest to $\bar{\mathbf{z}}$. Therefore, we can now write:

$$\hat{\mathbf{Z}} = D(E(\mathbf{Z})) = [\hat{Q}(\bar{\mathbf{z}}^{(1)}), \cdots, \hat{Q}(\bar{\mathbf{z}}^{(m)})] = \mathbf{C}[\hat{\phi}(\bar{\mathbf{z}}^{(1)}), \cdots, \hat{\phi}(\bar{\mathbf{z}}^{(m)})].$$

Now, instead of computing $\hat{\mathbf{Z}}$ via hard nearest neighbor assignments, we can approximate it with a smooth relaxation $\tilde{\mathbf{Z}} := \mathbf{C}[\phi(\bar{\mathbf{z}}^{(1)}), \cdots, \phi(\bar{\mathbf{z}}^{(m)})]$ by using the soft assignments instead of the hard assignments. Denoting the corresponding vector form by $\tilde{\mathbf{z}}$, this gives us a differentiable approximation $\tilde{F}$ of the quantized architecture $\hat{F}$, by replacing $\hat{\mathbf{z}}$ in the network with $\tilde{\mathbf{z}}$.

**Entropy estimation.** Using the soft assignments, we can similarly define a soft histogram, by summing up the partial assignments to each center instead of counting as in (4):

$$q_j := \frac{1}{mN} \sum_{i=1}^{N} \sum_{l=1}^{m} \phi_j(\bar{\mathbf{z}}_i^{(l)}).$$

This gives us a valid probability mass function $q = (q_1, \cdots, q_L)$, which is differentiable but converges to $p = (p_1, \cdots, p_L)$ as $\sigma \to \infty$.

We can now define the "soft entropy" as the cross entropy between $p$ and $q$:

$$\tilde{H}(\phi) := H(p, q) = -\sum_{j=1}^{L} p_j \log q_j = H(p) + D_{KL}(p\|q)$$

where $D_{KL}(p\|q) = \sum_j p_j \log(p_j/q_j)$ denotes the Kullback–Leibler divergence. Since $D_{KL}(p\|q) \geq 0$, this establishes $\tilde{H}(\phi)$ as an upper bound for $H(p)$, where equality is obtained when $p = q$.

We have therefore obtained a differentiable "soft entropy" loss (w.r.t. $q$), which is an upper bound on the sample entropy $H(p)$. Hence, we can indirectly minimize $H(p)$ by minimizing $\tilde{H}(\phi)$, treating the histogram probabilities of $p$ as constants for gradient computation. However, we note that while $q_j$ is additive over the training data and the symbol sequence, $\log(q_j)$ is not. This prevents the use of mini-batch gradient descent on $\tilde{H}(\phi)$, which can be an issue for large scale learning problems. In this case, we can instead re-define the soft entropy $\tilde{H}(\phi)$ as $H(q, p)$. As before, $\tilde{H}(\phi) \to H(p)$ as $\sigma \to \infty$, but $\tilde{H}(\phi)$ ceases to be an upper bound for $H(p)$. The benefit is that now $\tilde{H}(\phi)$ can be decomposed as

$$\tilde{H}(\phi) := H(q, p) = -\sum_{j=1}^{L} q_j \log p_j = -\sum_{i=1}^{N}\sum_{l=1}^{m}\sum_{j=1}^{L} \frac{1}{mN} \phi_j(\bar{\mathbf{z}}_i^{(l)}) \log p_j, \tag{8}$$

such that we get an additive loss over the samples $\mathbf{x}_i \in \mathcal{X}$ and the components $l \in [m]$.

**Soft-to-hard deterministic annealing.** Our soft assignment scheme gives us differentiable approximations $\tilde{F}$ and $\tilde{H}(\phi)$ of the discretized network $\hat{F}$ and the sample entropy $H(p)$, respectively. However, our objective is to learn network parameters $\mathbf{W}$ that minimize (6) when using the encoder and decoder with hard assignments, such that we obtain a compressible symbol stream $E(\mathbf{z})$ which we can compress using, *e.g.*, arithmetic coding [40].

To this end, we anneal $\sigma$ from some initial value $\sigma_0$ to infinity during training, such that the soft approximation gradually becomes a better approximation of the final hard quantization we will use. Choosing the annealing schedule is crucial as annealing too slowly may allow the network to invert the soft assignments (resulting in large weights), and annealing too fast leads to vanishing gradients too early, thereby preventing learning. In practice, one can either parametrize $\sigma$ as a function of the iteration, or tie it to an auxiliary target such as the difference between the network losses incurred by soft quantization and hard quantization (see Section 4 for details).

For a simple initialization of $\sigma_0$ and the centers $\mathcal{C}$, we can sample the centers from the set $\mathcal{Z} := \{\bar{\mathbf{z}}_i^{(l)} | i \in [N], l \in [m]\}$ and then cluster $\mathcal{Z}$ by minimizing the cluster energy $\sum_{\bar{\mathbf{z}} \in \mathcal{Z}} \|\bar{\mathbf{z}} - \tilde{Q}(\bar{\mathbf{z}})\|^2$ using SGD.

## 4 Image Compression

We now show how we can use our framework to realize a simple image compression system. For the architecture, we use a variant of the convolutional autoencoder proposed recently in [30] (see Appendix A.1 for details). We note that while we use the architecture of [30], we train it using our soft-to-hard entropy minimization method, which differs significantly from their approach, see below.

Our goal is to learn a compressible representation of the features in the bottleneck of the autoencoder. Because we do not expect the features from different bottleneck channels to be identically distributed, we model each channel's distribution with a different histogram and entropy loss, adding each entropy term to the total loss using the same $\beta$ parameter. To encode a channel into symbols, we separate the channel matrix into a sequence of $p_w \times p_h$-dimensional patches. These patches (vectorized) form the columns of $\mathbf{Z} \in \mathbb{R}^{d/m \times m}$, where $m = d/(p_w p_h)$, such that $\mathbf{Z}$ contains $m$ $(p_w p_h)$-dimensional points. Having $p_h$ or $p_w$ greater than one allows symbols to capture local correlations in the bottleneck, which is desirable since we model the symbols as i.i.d. random variables for entropy coding. At test time, the symbol encoder $E$ then determines the symbols in the channel by performing a nearest neighbor assignment over a set of $L$ centers $\mathcal{C} \subset \mathbb{R}^{p_w p_h}$, resulting in $\hat{\mathbf{Z}}$, as described above. During training we instead use the soft quantized $\tilde{\mathbf{Z}}$, also w.r.t. the centers $\mathcal{C}$.

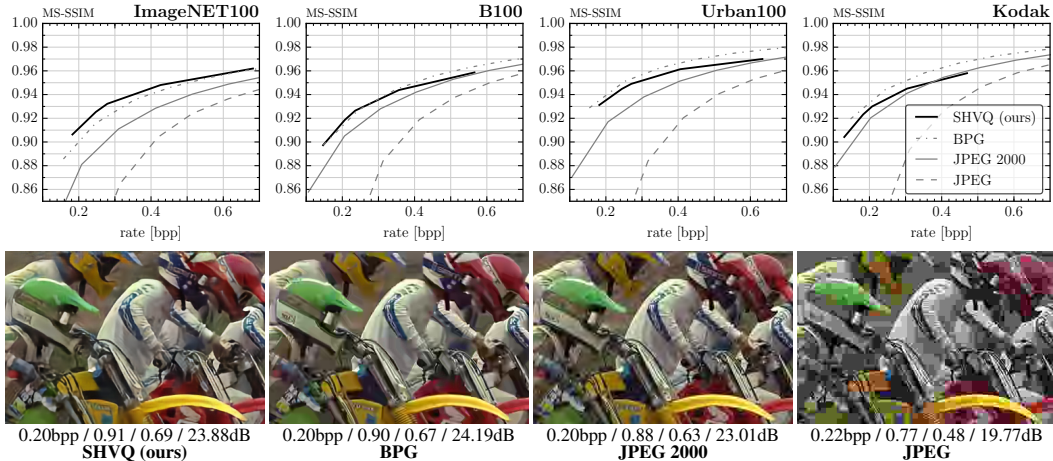

0.20bpp / 0.91 / 0.69 / 23.88dB    0.20bpp / 0.90 / 0.67 / 24.19dB    0.20bpp / 0.88 / 0.63 / 23.01dB    0.22bpp / 0.77 / 0.48 / 19.77dB
**SHVQ (ours)**          **BPG**          **JPEG 2000**          **JPEG**

Figure 1: Top: MS-SSIM as a function of rate for SHVQ (Ours), BPG, JPEG 2000, JPEG, for each data set. Bottom: A visual example from the Kodak data set along with rate / MS-SSIM / SSIM / PSNR.

We trained different models using Adam [17], see Appendix A.2. Our training set is composed similarly to that described in [4]. We used a subset of 90,000 images from ImageNET [9], which we downsampled by a factor 0.7 and trained on crops of $128 \times 128$ pixels, with a batch size of 15. To estimate the probability distribution $p$ for optimizing (8), we maintain a histogram over 5,000 images, which we update every 10 iterations with the images from the current batch. Details about other hyperparameters can be found in Appendix A.2.

The training of our autoencoder network takes place in two stages, where we move from an identity function in the bottleneck to hard quantization. In the first stage, we train the autoencoder without any quantization. Similar to [30] we gradually unfreeze the channels in the bottleneck during training (this gives a slight improvement over learning all channels jointly from the start). This yields an efficient weight initialization and enables us to then initialize $\sigma_0$ and $\mathcal{C}$ as described above. In the second stage, we minimize (6), jointly learning network weights and quantization levels. We anneal $\sigma$ by letting the gap between soft and hard quantization error go to zero as the number of iterations $t$ goes to infinity. Let $e_S = \|\tilde{F}(\mathbf{x}) - \mathbf{x}\|^2$ be the soft error, $e_H = \|\hat{F}(\mathbf{x}) - \mathbf{x}\|^2$ be the hard error. With $\text{gap}(t) = e_H - e_S$ we can denote the error between the actual the desired gap with $e_G(t) = \text{gap}(t) - T/(T+t) \, \text{gap}(0)$, such that the gap is halved after $T$ iterations. We update $\sigma$ according to $\sigma(t+1) = \sigma(t) + K_G \, e_G(t)$, where $\sigma(t)$ denotes $\sigma$ at iteration $t$. Fig. 3 in Appendix A.4 shows the evolution of the gap, soft and hard loss as sigma grows during training. We observed that both vector quantization and entropy loss lead to higher compression rates at a given reconstruction MSE compared to scalar quantization and training without entropy loss, respectively (see Appendix A.3 for details).

**Evaluation.** To evaluate the image compression performance of our Soft-to-Hard Vector Quantization Autoencoder (SHVQ) method we use four datasets, namely Kodak [2], B100 [31], Urban100 [14], ImageNET100 (100 randomly selected images from ImageNET [25]) and three standard quality measures, namely peak signal-to-noise ratio (PSNR), structural similarity index (SSIM) [37], and multi-scale SSIM (MS-SSIM), see Appendix A.5 for details. We compare our SHVQ with the standard JPEG, JPEG 2000, and BPG [1], focusing on compression rates $< 1$ bits per pixel (bpp) (*i.e.*, the regime where traditional integral transform-based compression algorithms are most challenged). As shown in Fig. 1, for high compression rates ($< 0.4$ bpp), our SHVQ outperforms JPEG and JPEG 2000 in terms of MS-SSIM and is competitive with BPG. A similar trend can be observed for SSIM (see Fig. 4 in Appendix A.6 for plots of SSIM and PSNR as a function of bpp). SHVQ performs best on ImageNET100 and is most challenged on Kodak when compared with JPEG 2000. Visually, SHVQ-compressed images have fewer artifacts than those compressed by JPEG 2000 (see Fig. 1, and Fig. 5–12 in Appendix A.7).

**Related methods and discussion.** JPEG 2000 [29] uses wavelet-based transformations and adaptive EBCOT coding. BPG [1], based on a subset of the HEVC video compression standard, is the

| METHOD | ACC [%] | COMP. RATIO |
|---|---|---|
| ORIGINAL MODEL | 92.6 | 1.00 |
| PRUNING + FT. + INDEX CODING + H. CODING [12] | 92.6 | 4.52 |
| PRUNING + FT. + K-MEANS + FT. + I.C. + H.C. [11] | 92.6 | 18.25 |
| PRUNING + FT. + HESSIAN-WEIGHTED K-MEANS + FT. + I.C. + H.C. | 92.7 | 20.51 |
| PRUNING + FT. + UNIFORM QUANTIZATION + FT. + I.C. + H.C. | 92.7 | 22.17 |
| PRUNING + FT. + ITERATIVE ECSQ + FT. + I.C. + H.C. | 92.7 | 21.01 |
| SOFT-TO-HARD ANNEALING + FT. + H. CODING (OURS) | 92.1 | 19.15 |
| SOFT-TO-HARD ANNEALING + FT. + A. CODING (OURS) | 92.1 | 20.15 |

Table 1: Accuracies and compression factors for different DNN compression techniques, using a 32-layer ResNet on CIFAR-10. FT. denotes fine-tuning, IC. denotes index coding and H.C. and A.C. denote Huffman and arithmetic coding, respectively. The pruning based results are from [6].

current state-of-the art for image compression. It uses context-adaptive binary arithmetic coding (CABAC) [21].

The recent works of [30, 5] also showed competitive performance with JPEG 2000. While we use the architecture of [30], there are stark differences between the works, summarized

| | SHVQ (ours) | Theis *et al.* [30] |
|---|---|---|
| Quantization | vector quantization | rounding to integers |
| Backpropagation | grad. of soft relaxation | grad. of identity mapping |
| Entropy estimation | (soft) histogram | Gaussian scale mixtures |
| Training material | ImageNET | high quality Flickr images |
| Operating points | single model | ensemble |

in the inset table. The work of [5] build a deep model using multiple generalized divisive normalization (GDN) layers and their inverses (IGDN), which are specialized layers designed to capture local joint statistics of natural images. Furthermore, they model marginals for entropy estimation using linear splines and also use CABAC[21] coding. Concurrent to our work, the method of [16] builds on the architecture proposed in [33], and shows that impressive performance in terms of the MS-SSIM metric can be obtained by incorporating it into the optimization (instead of just minimizing the MSE).

In contrast to the domain-specific techniques adopted by these state-of-the-art methods, our framework for learning compressible representation can realize a competitive image compression system, only using a convolutional autoencoder and simple entropy coding.

## 5 DNN Compression

For DNN compression, we investigate the ResNet [13] architecture for image classification. We adopt the same setting as [6] and consider a 32-layer architecture trained for CIFAR-10 [18]. As in [6], our goal is to learn a compressible representation for all 464,154 trainable parameters of the model.

We concatenate the parameters into a vector $\mathbf{W} \in \mathbb{R}^{464,154}$ and employ scalar quantization ($m = d$), such that $\mathbf{Z}^T = \mathbf{z} = \mathbf{W}$. We started from the pre-trained original model, which obtains a $92.6\%$ accuracy on the test set. We implemented the entropy minimization by using $L = 75$ centers and chose $\beta = 0.1$ such that the converged entropy would give a compression factor $\approx 20$, *i.e.*, giving $\approx 32/20 = 1.6$ bits per weight. The training was performed with the same learning parameters as the original model was trained with (SGD with momentum $0.9$). The annealing schedule used was a simple exponential one, $\sigma(t + 1) = 1.001 \cdot \sigma(t)$ with $\sigma(0) = 0.4$. After 4 epochs of training, when $\sigma(t)$ has increased by a factor $\approx 20$, we switched to hard assignments and continued fine-tuning at a $10\times$ lower learning rate. [2] Adhering to the benchmark of [6, 12, 11], we obtain the compression factor by dividing the bit cost of storing the uncompressed weights as floats ($464,154 \times 32$ bits) with the total encoding cost of compressed weights (*i.e.*, $L \times 32$ bits for the centers plus the size of the compressed index stream).

Our compressible model achieves a comparable test accuracy of $92.1\%$ while compressing the DNN by a factor $19.15$ with Huffman and $20.15$ using arithmetic coding. Table 1 compares our results with state-of-the-art approaches reported by [6]. We note that while the top methods from the literature also achieve accuracies above $92\%$ and compression factors above $20\times$, they employ a considerable amount of hand-designed steps, such as pruning, retraining, various types of weight clustering, special encoding of the sparse weight matrices into an index-difference based format and then finally use

entropy coding. In contrast, we directly minimize the entropy of the weights in the training, obtaining a highly compressible representation using standard entropy coding.

In Fig. 13 in Appendix A.8, we show how the sample entropy $H(p)$ decays and the index histograms develop during training, as the network learns to condense most of the weights to a couple of centers when optimizing (6). In contrast, the methods of [12, 11, 6] manually impose 0 as the most frequent center by pruning $\approx 80\%$ of the network weights. We note that the recent works by [34] also manages to tackle the problem in a single training procedure, using the minimum description length principle. In contrast to our framework, they take a Bayesian perspective and rely on a parametric assumption on the symbol distribution.

## 6   Conclusions

In this paper we proposed a unified framework for end-to-end learning of compressed representations for deep architectures. By training with a soft-to-hard annealing scheme, gradually transferring from a soft relaxation of the sample entropy and network discretization process to the actual non-differentiable quantization process, we manage to optimize the rate distortion trade-off between the original network loss and the entropy. Our framework can elegantly capture diverse compression tasks, obtaining results competitive with state-of-the-art for both image compression as well as DNN compression. The simplicity of our approach opens up various directions for future work, since our framework can be easily adapted for other tasks where a compressible representation is desired.

## Acknowledgments

This work was supported by EUs Horizon 2020 programme under grant agreement No 687757 – REPLICATE, by NVIDIA Corporation through the Academic Hardware Grant, by ETH Zurich, and by Armasuisse.

## Footnotes

[1]In fact, from [8], Thm. 2.6.6, it follows that if the histogram estimates $p_j$ are exact, (5) is an upper bound for the true $H(E(\mathsf{Z}))$ (i.e., without the i.i.d. assumption).

[2] We switch to hard assignments since we can get large gradients for weights that are equally close to two centers as $\tilde{Q}$ converges to hard nearest neighbor assignments. One could also employ simple gradient clipping.

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
