[Supplementary Material]

# A    Soft-to-Hard Vector Quantization for End-to-End Learning Compressible Representations – Appendix

This is the supplementary material for the NIPS 2017 published paper:

**Title:** Soft-to-Hard Vector Quantization for End-to-End Learning Compressible Representations

**Authors:** Eirikur Agustsson, Fabian Mentzer, Michael Tschannen, Lukas Cavigelli, Radu Timofte, Luca Benini, Luc Van Gool

## A.1    Architecture

We rely on a variant of the compressive autoencoder proposed recently in [30], using convolutional neural networks for the image encoder and image decoder [3]. The first two convolutional layers in the image encoder each downsample the input image by a factor 2 and collectively increase the number of channels from 3 to 128. This is followed by three residual blocks, each with 128 filters. Another convolutional layer then downsamples again by a factor 2 and decreases the number of channels to $c$, where $c$ is a hyperparameter ([30] use 64 and 96 channels). For a $w \times h$-dimensional input image, the output of the image encoder is the $w/8 \times h/8 \times c$-dimensional "bottleneck tensor".

The image decoder then mirrors the image encoder, using upsampling instead of downsampling, and deconvolutions instead of convolutions, mapping the bottleneck tensor into a $w \times h$-dimensional output image. In contrast to the "subpixel" layers [26, 27] used in [30], we use standard deconvolutions for simplicity.

## A.2    Hyperparameters

We do vector quantization to $L = 1000$ centers, using $(p_w, p_h) = (2, 2)$, *i.e.*, $m = d/(2 \cdot 2)$. We trained different combinations of $\beta$ and $c$ to explore different rate-distortion tradeoffs (measuring distortion in MSE). As $\beta$ controls to which extent the network minimizes entropy, $\beta$ directly controls bpp (see top left plot in Fig. 3). We evaluated all pairs $(c, \beta)$ with $c \in \{8, 16, 32, 48\}$ and $m\beta \in \{1e^{-4}, \ldots, 9e^{-4}\}$, and selected 5 representative pairs (models) with average bpps roughly corresponding to uniformly spread points in the interval $[0.1, 0.8]$ bpp. This defines a "quality index" for our model family, analogous to the JPEG quality factor.

We experimented with the other training parameters on a setup with $c = 32$, which we chose as follows. In the first stage we train for $250k$ iterations using a learning rate of $1e^{-4}$. In the second stage, we use an annealing schedule with $T = 50k, K_G = 100$, over $800k$ iterations using a learning rate of $1e^{-5}$. In both stages, we use a weak $l_2$ regularizer over all learnable parameters, with $\lambda = 1e^{-12}$.

## A.3    Effect of Vector Quantization and Entropy Loss

Figure 2: PSNR on ImageNET100 as a function of the rate for $2 \times 2$-dimensional centers (Vector), for $1 \times 1$-dimensional centers (Scalar), and for $2 \times 2$-dimensional centers without entropy loss ($\beta = 0$). JPEG is included for reference.

To investigate the effect of vector quantization, we trained models as described in Section 4, but instead of using vector quantization, we set $L = 6$ and quantized to $1 \times 1$-dimensional (scalar) centers,

*i.e.*, $(p_h, p_w) = (1,1), m = d$. Again, we chose 5 representative pairs $(c, \beta)$. We chose $L = 6$ to get approximately the same number of unique symbol assignments as for $2 \times 2$ patches, *i.e.*, $6^4 \approx 1000$.

To investigate the effect of the entropy loss, we trained models using $2 \times 2$ centers for $c \in \{8, 16, 32, 48\}$ (as described above), but used $\beta = 0$.

Fig. 2 shows how both vector quantization and entropy loss lead to higher compression rates at a given reconstruction MSE compared to scalar quantization and training without entropy loss, respectively.

## A.4 Effect of Annealing

Figure 3: Entropy loss for three $\beta$ values, soft and hard PSNR, as well as gap$(t)$ and $\sigma$ as a function of the iteration $t$.

## A.5 Data Sets and Quality Measure Details

**Kodak** [2] is the most frequently employed dataset for analizing image compression performance in recent years. It contains 24 color $768 \times 512$ images covering a variety of subjects, locations and lighting conditions.

**B100** [31] is a set of 100 content diverse color $481 \times 321$ test images from the Berkeley Segmentation Dataset [22].

**Urban100** [14] has 100 color images selected from Flickr with labels such as urban, city, architecture, and structure. The images are larger than those from B100 or Kodak, in that the longer side of an image is always bigger than 992 pixels. Both B100 and Urban100 are commonly used to evaluate image super-resolution methods.

**ImageNET100** contains 100 images randomly selected by us from ImageNET [25], also downsampled and cropped, see above.

**Quality measures.**    PSNR (peak signal-to-noise ratio) is a standard measure in direct monotonous relation with the mean square error (MSE) computed between two signals. SSIM and MS-SSIM are the structural similarity index [37] and its multi-scale SSIM computed variant [36] proposed to measure the similarity of two images. They correlate better with human perception than PSNR.

We compute quantitative similarity scores between each compressed image and the corresponding uncompressed image and average them over whole datasets of images. For comparison with JPEG we used libjpeg[4], for JPEG 2000 we used the Kakadu implementation[5], subtracting in both cases the size of the header from the file size to compute the compression rate. For comparison with BPG we used the reference implementation[6] and used the value reported in the `picture_data_length` header field as file size.

## A.6    Image Compression Performance

Figure 4: Average MS-SSIM, SSIM, and PSNR as a function of the rate for the ImageNET100, Urban100, B100 and Kodak datasets.

## A.7    Image Compression Visual Examples

Fig. 5–12 show the output of compressing the first four images of each of the four datasets with our method, BPG, JPEG, and JPEG 2000, at low bitrates.

**SHVQ (ours)** 0.15bpp / 0.92 / 0.75 / 27.75dB      0.15bpp / 0.93 / 0.78 / 29.31dB **BPG**

**JPEG** 0.16bpp / 0.76 / 0.58 / 21.38dB      0.15bpp / 0.91 / 0.74 / 27.87dB **JPEG 2000**

**SHVQ (ours)** 0.20bpp / 0.92 / 0.72 / 24.13dB      0.21bpp / 0.93 / 0.74 / 24.93dB **BPG**

**JPEG** 0.21bpp / 0.83 / 0.58 / 20.28dB      0.21bpp / 0.91 / 0.68 / 23.60dB **JPEG 2000**

Figure 5: Our SHVQ vs. BPG, JPEG and JPEG 2000 on the first and second image of the ImageNET100 dataset, along with bit rate / MS-SSIM / SSIM / PSNR.

**SHVQ (ours)** 0.21bpp / 0.91 / 0.68 / 23.57dB     0.23bpp / 0.92 / 0.68 / 24.35dB **BPG**

**JPEG** 0.21bpp / 0.80 / 0.51 / 20.14dB     0.22bpp / 0.90 / 0.65 / 23.51dB **JPEG 2000**

**SHVQ (ours)** 0.24bpp / 0.92 / 0.67 / 21.08dB     0.27bpp / 0.91 / 0.66 / 21.73dB **BPG**

**JPEG** 0.26bpp / 0.79 / 0.48 / 18.25dB     0.24bpp / 0.89 / 0.61 / 20.72dB **JPEG 2000**

Figure 6: Our SHVQ vs. BPG, JPEG and JPEG 2000 on the third and forth image of the ImageNET100 dataset, along with bit rate / MS-SSIM / SSIM / PSNR.

**SHVQ (ours)** 0.17bpp / 0.88 / 0.54 / 22.69dB          0.19bpp / 0.88 / 0.54 / 23.13dB **BPG**

**JPEG** 0.18bpp / 0.74 / 0.36 / 19.54dB          0.18bpp / 0.87 / 0.52 / 22.32dB **JPEG 2000**

**SHVQ (ours)** 0.12bpp / 0.93 / 0.77 / 26.69dB          0.12bpp / 0.94 / 0.80 / 27.71dB **BPG**

**JPEG** 0.16bpp / 0.78 / 0.63 / 22.01dB          0.13bpp / 0.92 / 0.76 / 26.43dB **JPEG 2000**

Figure 7: Our SHVQ vs. BPG, JPEG and JPEG 2000 on the first and second image of the B100 dataset, along with bit rate / MS-SSIM / SSIM / PSNR.

**SHVQ (ours)** 0.14bpp / 0.93 / 0.79 / 25.96dB          0.14bpp / 0.93 / 0.80 / 27.17dB **BPG**

**JPEG** 0.17bpp / 0.76 / 0.63 / 21.11dB          0.15bpp / 0.92 / 0.77 / 26.24dB **JPEG 2000**

**SHVQ (ours)** 0.11bpp / 0.91 / 0.76 / 27.59dB          0.10bpp / 0.91 / 0.76 / 28.04dB **BPG**

**JPEG** 0.15bpp / 0.72 / 0.53 / 22.34dB          0.12bpp / 0.91 / 0.75 / 27.81dB **JPEG 2000**

Figure 8: Our SHVQ vs. BPG, JPEG and JPEG 2000 on the third and forth image of the B100 dataset, along with bit rate / MS-SSIM / SSIM / PSNR.

**SHVQ (ours)** 0.14bpp / 0.92 / 0.68 / 25.95dB        0.13bpp / 0.92 / 0.69 / 27.00dB **BPG**

**JPEG** 0.18bpp / 0.79 / 0.52 / 21.61dB        0.14bpp / 0.89 / 0.63 / 25.28dB **JPEG 2000**

**SHVQ (ours)** 0.17bpp / 0.95 / 0.79 / 25.28dB        0.17bpp / 0.95 / 0.80 / 26.05dB **BPG**

**JPEG** 0.19bpp / 0.82 / 0.54 / 20.80dB        0.17bpp / 0.93 / 0.73 / 24.57dB **JPEG 2000**

Figure 9: Our SHVQ vs. BPG, JPEG and JPEG 2000 on the first and second image of the Urban100 dataset, along with bit rate / MS-SSIM / SSIM / PSNR.

**SHVQ (ours)** 0.19bpp / 0.91 / 0.66 / 22.85dB · 0.20bpp / 0.90 / 0.65 / 23.28dB **BPG**

**JPEG** 0.21bpp / 0.76 / 0.43 / 19.25dB · 0.20bpp / 0.88 / 0.59 / 22.14dB **JPEG 2000**

**SHVQ (ours)** 0.22bpp / 0.96 / 0.83 / 24.74dB · 0.22bpp / 0.97 / 0.85 / 26.30dB **BPG**

**JPEG** 0.24bpp / 0.86 / 0.62 / 20.13dB · 0.23bpp / 0.95 / 0.79 / 24.37dB **JPEG 2000**

Figure 10: Our SHVQ vs. BPG, JPEG and JPEG 2000 on the third and forth image of the Urban100 dataset, along with bit rate / MS-SSIM / SSIM / PSNR.

**SHVQ (ours)** 0.16bpp / 0.89 / 0.63 / 24.42dB

0.16bpp / 0.90 / 0.64 / 24.97dB **BPG**

**JPEG** 0.18bpp / 0.70 / 0.42 / 19.90dB

0.17bpp / 0.87 / 0.59 / 24.02dB **JPEG 2000**

**SHVQ (ours)** 0.12bpp / 0.88 / 0.72 / 28.76dB

0.12bpp / 0.91 / 0.78 / 30.99dB **BPG**

**JPEG** 0.14bpp / 0.58 / 0.49 / 21.73dB

0.12bpp / 0.89 / 0.74 / 30.06dB **JPEG 2000**

Figure 11: Our SHVQ vs. BPG, JPEG and JPEG 2000 on the first and second image of the Kodak dataset, along with bit rate / MS-SSIM / SSIM / PSNR.

**SHVQ (ours)** 0.10bpp / 0.93 / 0.82 / 29.81dB      0.11bpp / 0.95 / 0.87 / 32.14dB **BPG**

**JPEG** 0.14bpp / 0.76 / 0.67 / 22.68dB      0.10bpp / 0.93 / 0.82 / 30.42dB **JPEG 2000**

**SHVQ (ours)** 0.11bpp / 0.89 / 0.73 / 28.36dB      0.12bpp / 0.92 / 0.78 / 30.14dB **BPG**

**JPEG** 0.14bpp / 0.64 / 0.54 / 21.21dB      0.11bpp / 0.90 / 0.75 / 29.19dB **JPEG 2000**

Figure 12: Our SHVQ vs. BPG, JPEG and JPEG 2000 on the third and forth image of the Kodak dataset, along with bit rate / MS-SSIM / SSIM / PSNR.

## A.8 DNN Compression: Entropy and Histogram Evolution

Figure 13: We show how the sample entropy $H(p)$ decays during training, due to the entropy loss term in (6), and corresponding index histograms at three time instants. Top left: Evolution of the sample entropy $H(p)$. Top right: the histogram for the entropy $H = 4.07$ at $t = 216$. Bottom left and right: the corresponding sample histogram when $H(p)$ reaches 2.90 bits per weight at $t = 475$ and the final histogram for $H(p) = 1.58$ bits per weight at $t = 520$.

## Footnotes

[3]We note that the image encoder (decoder) refers to the left (right) part of the autoencoder, which encodes (decodes) the data to (from) the bottleneck (not to be confused with the symbol encoder (decoder) in Section 3).

[4] `http://libjpeg.sourceforge.net/`

[5] `http://kakadusoftware.com/`

[6] `https://bellard.org/bpg/`