[Reviews · NeurIPS 2017]

Reviewer 1



The paper aims to produce compressible representations. The paper introduces two new tools: 1) Differentiable vector quantization. The hard assignment to the nearest center vector is approximated by soft softmax weights. 2) Entropy estimation for the marginal code distribution. The entropy is estimated by counting the occurrences of code symbols. The counting is done over a subset of the training data. The vector quantization works better than scalar quantization. Some minor cons: - The entropy estimation would be expensive to do exactly. So the computation is approximated only on a subset of the training data. - The method becomes less simple, if considering the autoencoder pretraining, the minimization of the cluster energy and the softmax temperature annealing. - It is not clear how to prevent the network to invert the soft assignments. Suggestions for clarity improvements: - When reading section 3, I wondered how p was computed. The estimation was explained only in the experimental Section 4. - The SHA abbreviation collides with the existing Secure Hash Algorithms (SHA). SHAE may be a better name (Soft-to-Hard AutoEncoder). Questions about evaluation: - How is the bitrate computed? Is the P(E(Z)) prior frozen after training? And is the number of bits computed by -logP(E(z_{testImage}))? I expect that the test images may have higher coding cost than the training images. Update: I have read the rebuttal.

Reviewer 2



The paper proposes an end-to-end approach for compressing neural networks. Instead of learning the network and then reducing it, they learn a network such that the parameters are compressible. They obtain results competitive with state-of-the-art while having a data-dependent and end-to-end approach, which is a very nice result. I particularly enjoyed reading this paper, I found it very clear and well motivated. The comparison to the current literature in that area is very thorough and the mathematical motivations are well formulated. The softmax with temperature sigma is a simple trick for soft-to-hard encoding that elegantly solves the non-differentiability problem in quantization and allows to control the smoothness of the assignment. I found the soft assignments and entropy estimation to be elegant solutions to the well-defined problem the authors described in section 3.1. Though I am not an expert in that field, I found this work very nice and it seems to be a significant step towards having end-to-end approaches for compressing neural networks, which look promising compared to previous approaches. This can have a large impact considering the increasing use of neural networks on mobile or embedded devices with limited memory.

Reviewer 3



This paper proposed a unified end-to-end framework for training neural networks to get compressible representations or weights. The proposed method encourages compressibility by minimizing the entropy of representation or weight. Because the original entropy-based objective cannot be minimized directly, the authors instead relax the objective and use discretized approximations. The proposed method also use an annealing schedule to start from a soft discretization to gradually transform to a hard discretization. My main concerns with this paper are on experiments and comparisons. The authors performed two sets of experiments, one on compressing representations of image and the other on compressing classifier weights. For the image compression experiment, the proposed method offers visually appealing results but it is only compared to BPG and JPEG. I suggest the authors also include comparisons against other neural network based approaches [1]. In the weight compression experiment, the proposed method does not show advantage over previous state-of-the-art. The authors also noted this but argued that the proposed method is trained end-to-end while the previous methods have many hand designed steps. However, I would expect integrated end-to-end training to improve performance instead of hindering it. I suggest the authors investigate more and improve their method to fully realize the benefit of end-to-end training. [1] Full Resolution Image Compression with Recurrent Neural Networks